# What Is behind the Correlation Analysis of Diarrheagenic *E. coli* Pathotypes?

**DOI:** 10.3390/biology11071004

**Published:** 2022-07-02

**Authors:** Mahmoud M. Bendary, Marwa I. Abd El-Hamid, Majid Alhomrani, Abdulhakeem S. Alamri, Rana Elshimy, Rasha A. Mosbah, Mosa M. Bahnass, Nasreen N. Omar, Mohammad M. Al-Sanea, Arwa R. Elmanakhly, Nesreen A. Safwat, Walaa A. Alshareef

**Affiliations:** 1Department of Microbiology and Immunology, Faculty of Pharmacy, Port Said University, Port Said 42511, Egypt; 2Department of Microbiology, Faculty of Veterinary Medicine, Zagazig University, Zagazig 44511, Egypt; mero_micro2006@yahoo.com; 3Department of Clinical Laboratories Sciences, The Faculty of Applied Medical Science, Taif University, Taif 26432, Saudi Arabia; m.alhomarani@tu.edu.sa (M.A.); a.alamri@tu.edu.sa (A.S.A.); 4Centre of Biomedical Science Research (CBSR), Deanship of Scientific Research, Taif University, Taif 26432, Saudi Arabia; 5Department of Microbiology and Immunology, Egyptian Drug Authority, Giza 12511, Egypt; rokashimy@yahoo.com; 6Department of Microbiology and Immunology, Faculty of Pharmacy, Alhram Canadian University, Giza 12511, Egypt; 7Infection Control Unit, Zagazig University Hospital, Zagazig 44511, Egypt; rashamosbah1@yahoo.com; 8Department of Animal Medicine (Infectious Disease), Faculty of Veterinary Medicine, Zagazig University, Zagazig 44511, Egypt; midomosa925@gmail.com; 9Department of Biochemistry, Faculty of Pharmacy, Modern University for Technology and Information, Cairo 19448, Egypt; nasreen.nabil@pharm.mti.edu.eg; 10Pharmaceutical Chemistry Department, College of Pharmacy, Jouf University, Sakaka 72314, Saudi Arabia; 11Department of Microbiology and Immunology, Faculty of Pharmacy, Modern University for Technology and Information, Cairo 11559, Egypt; arwa.ramadan82@gmail.com (A.R.E.); nesreensafwat@hotmail.com (N.A.S.); 12Department of Microbiology and Immunology, Faculty of Pharmacy, October 6 University, 6 October 12566, Egypt; lolowation@gmail.com

**Keywords:** *E. coli*, pathotypes, hosts, correlation, clonality

## Abstract

**Simple Summary:**

To date, despite the efforts made to monitor the wide spread of resistant pathogens, especially multidrug-resistant (MDR) diarrheagenic *E. coli*, there are limitations in the correlation analysis for these pathogens worldwide. Therefore, it seems important and so timely to assess the *E. coli* pathotypes and their correlations with hosts, antimicrobial resistance, virulence gene profiles, and serotypes. Our promising results gave a clear indication for the epidemic situation of diarrheagenic *E. coli* (DEC) in Egypt and suggested that restricted recommendations and a search for novel alternative therapies are urgently needed due to the wide spread of MDR and multi-virulent *E. coli* strains in addition to their heterogeneous nature. This study can be implemented in the infection control guidance with enhanced protocols to hinder the spread of MDR *E. coli* pathotypes in Egyptian hospitals.

**Abstract:**

The treatment failure recorded among patients and animals infected with diarrheagenic *Escherichia coli* (DEC) was increased due to the presence of specific virulence markers among these strains. These markers were used to classify DEC into several pathotypes. We analyzed the correlations between DEC pathotypes and antimicrobial resistances, the existence of virulence genes, serotypes, and hosts. The ETEC pathotype was detected with a high prevalence rate (25%). Moreover, the ETEC and EPEC pathotypes were highly associated with human infections in contrast to the EIEC and EAEC phenotypes, which were commonly recognized among animal isolates. Interestingly, the antimicrobial resistance was affected by *E. coli* pathotypes. With the exception of EIEC and STEC, imipenem represented the most effective antibiotic against the other pathotypes. There were fixed correlations between the DEC pathotypes and the presence of virulence markers and hosts; meanwhile, their correlation with serotypes was variable. Additionally, the vast majority of our isolates were highly diverse, based on both phenotypic and ERIC molecular typing techniques. Our promising results gave a clear indication for the heterogeneity and weak clonality of DEC pathotypes in Egypt, which can be utilized in the evaluation of the current therapeutic protocols and infection control guidelines.

## 1. Introduction

*Escherichia coli* (*E. coli*) strains are normal inhabitants of both animal and human gastrointestinal tracts, and they are among the bacterial species that are most frequently recovered from stool cultures [1]. When *E. coli* strains obtain precise genetic materials, they become pathogenic; these pathogenic *E. coli* clones are among the most virulent enteric bacterial pathogens [2]. Taken together, most *E. coli* strains cause serious health problems for both humans and animals as they can cause serious intestinal as well as extraintestinal infections. Therefore, next-generation therapies, in addition to discovering and developing novel antimicrobial agents, are urgently needed [3,4]. Diarrheagenic *E. coli* (DEC) are among the most abundant bacterial pathogens that cause gastroenteritis worldwide [5,6]. 

Recently, several DEC pathotypes were relatively defined. Six categories of DEC were documented based to their serotypes, particular virulence features, and various clinical and epidemiological features. They include enteropathogenic *E. coli* (EPEC), enteroinvasive *E. coli* (EIEC), shiga-toxin-producing *E. coli* (STEC), enterotoxigenic *E. coli* (ETEC), enterohaemorrhagic *E. coli* (EHEC), and enteroaggregative *E. coli* (EAEC) [6]. Numerous virulence-associated factors contribute to the pathogenicity of *E. coli* pathotypes since they help in the colonization of these microorganisms to the host surfaces, the invasion of host tissues, the avoidance of the host defense mechanisms, and the stimulation of the inflammatory responses with a consequence of causing clinical diseases [7].

In the epidemiological research, determining the genetic relationships between the clinically important *E. coli* species and tracking their infection sources are essential for assessing the emergent risk to public health and for enabling more targeted approaches to reducing the spread of pathogenic *E. coli* strains [8]. Additionally, the correlation analysis between DEC pathotypes and antimicrobial resistance as well as virulence profiles may help physicians to avoid treatment failure. The choice of antimicrobial therapies relies on the type of DEC in addition to its virulence and resistance profiles. Therefore, success in DEC treatment depends on the accurate and rapid characterization of DEC pathotypes. For the accurate control of DEC-related diarrhea, especially in children, the pathotyping and phenotypic or genotypic testing should usually be incorporated in medical settings [9,10]. Recently, various genotypic methods have been developed as appropriate tools for the molecular epidemiological investigations [11]. Among the PCR-based methods, enterobacterial repetitive intergenic consensus (ERIC) PCR is considered a quick, sharp, simple, and cost-effective genotyping approach used for characterizing the diversity of *E. coli* strains.

Therefore, the current study was planned to clarify the occurrence of DEC as a major cause of diarrhea in both humans and animals and to additionally characterize their pathotypes and genetic relatedness. We also employed correlation analyses between pathotypes and host, antimicrobial resistance, serotypes, and virulence profiles as well as the molecular typing of DEC using an ERIC-PCR technique to clarify the epidemic situation of the circulating pathotypes in Egypt. 

## 2. Materials and Methods

### 2.1. Ethical Statement

The investigated isolates were kindly obtained from the laboratories of microbiology at the Faculty of Veterinary Medicine, Zagazig University, and the Faculty of Pharmacy, Port-Said University. Therefore, the participants’ informed consent and the ethical approval for performing this work were not required.

### 2.2. Phenotypic and Genotypic Confirmation of DEC Isolates

This is a retrospective study carried out on 140 (80 human and 60 cow and equine) DEC isolates, which were verified via using API 20E strips (BioMérieux, Mary l’Etoile, France) and applying the molecular identification of the *16S rRNA* gene [12].

### 2.3. Molecular Confirmation of DEC Pathotypes 

The definition of the molecularly characterized DEC isolates into pathotypes was carried out via PCR assays using particular primer sets for the detection of the appropriate virulence genes of the EPEC, STEC, EAEC, EIEC, EHEC, and ETEC pathotypes. The DEC isolates that harbored three or more virulence genes were categorized as multi-virulent [13]. All PCR amplification assays were performed in a total reaction mixture of 25 μL comprising 12.5 μL of 2X Dream*Taq*^TM^ Green PCR Master Mix (Fermentas, Hanover, Germany), 0.1 μL of 100 pmoL of each primer (Sigma-Aldrich, St. Louis, MO, USA), 2 μL of DEC genomic DNA, and DNase/RNase-free water (up to 25 μL). All PCR amplifications were carried out on a PTC-100^TM^ thermal cycler (Waltham, MA, USA), adopting the previously mentioned thermal cycling settings [14,15,16,17,18,19,20,21,22,23,24,25]. The used primer sequences and the amplicon sizes are described in Table 1. The amplified PCR products were verified by electrophoresis on 1.5% agarose gel (Sigma-Aldrich, St. Louis, MO, USA) after staining with 0.5 μg/mL ethidium bromide (Sigma-Aldrich, St. Louis, MO, USA) and were then visualized and immediately photographed over a UV transilluminator (Spectroline, Millipore, Sigma-Aldrich, St. Louis, MO, USA). A 100 bp DNA ladder (Fermentas, Hanover, Germany) that was included as a molecular marker was utilized to define the PCR fragment sizes. All PCR assays were repeated three times with appropriate PCR positive controls (DNA templates of previously identified DEC pathotypes) and no template/negative control. All positive DEC control strains were kindly obtained by the Animal Health Research Institute, Egypt. 

### 2.4. Serotyping

The serotyping of the investigated DEC isolates was carried out via agglutination tests with both polyvalent and monovalent O-specific antisera according to the manufacturers’ recommendations (Test Sera Enteroclon, Anti-Coli, Berlin, Germany). 

### 2.5. Antimicrobial Susceptibility Testing

Determining the antimicrobial susceptibility patterns of the tested DEC isolates was carried out via the standardized Kirby–Bauer disc diffusion technique using various antimicrobial discs and Muller Hinton agar following the Clinical and Laboratory Standards Institute (CLSI) guidelines [26]. The antimicrobial agents that were used are listed in Table 2. The disc susceptibility patterns were confirmed using the broth microdilution method [27]. The definition of MDR isolates was proposed as a non-susceptibility to at least one antimicrobial agent in three or more different classes [28].

### 2.6. Enterobacterial Repetitive Intergenic Consensus Polymerase Chain Reaction (ERIC-PCR) for DNA Fingerprinting of DEC 

For achieving the genomic fingerprinting of DEC isolates via ERIC-PCR, one pair of primer sets (ERIC1: 5-ATGTAAGCTCCTGGGGATTCAC-3′ and ERIC2: 5′-AAGTAAGTGACTGGGGTGAGCG-3′) was used. The amplification reactions comprised an initial denaturation step (95 °C, 2 min), then 35 cycles of denaturation (92 °C, 30 s), annealing (50 °C, 1 min), and extension (65 °C, 8 min), and finally an extension step (65 °C, 8 min) [29]. The ERIC-PCR fragments were examined via 1% gel electrophoresis with the 100-base-pair DNA ladder (Fermentas, Litvany) being used as a molecular weight size marker. The banding patterns and the amplicon sizes were used in the dendrogram construction. The 1 and 0 scores were attributed to the presence and absence of ERIC-PCR bands, respectively. The R program was used for constructing the computerized dendrograms. 

### 2.7. Statistical Analysis

A two-way analysis of variance (ANOVA) without replication was used to detect the significant variations in the of resistance levels of the investigated isolates against the antimicrobials that were used. A *p* value less than 0.05 was a typical statistically significant test result. All discriminatory power (DI-value) and statistical and correlation analyses were conducted via the R packages *corrplot*, *heatmaply*, *hmisc*, and *ggpubr* and GraphPad Prism, version 6, provided by GraphPad Software Inc., San Diego, CA, USA. The correlation coefficient (*r*) was determined to estimate the strengths of the associations between the pathotypes and the antimicrobial resistance, virulence profiles, serotypes, and hosts using the mentioned R packages. The *r*-values of 0:0.5 and 0.5:1 reflect weaker and stronger positive correlations, respectively; meanwhile, weaker and stronger negative correlations were detected when the *r*-values were equal to 0:−0.5 and 0.5:−1, respectively.

## 3. Results

### 3.1. Characterization and Virulence-Associated Features of DEC Pathotypes

All our DEC isolates (140) were confirmed by both the API 20E identification system and PCR detection of the *16SrRNA* gene. Considering the PCR amplification of DEC virulence genes under study, five pathotypes were detected among all human and animal isolates, including EPEC, EAEC, EIEC, STEC, and ETEC (Figure 1). Considering the prevalence rates of all DEC pathotypes, ETEC followed by EAEC represented the most common ones (25% and 22.1%, respectively); meanwhile, EIEC had the lowest prevalence rate (17.1%), as shown in Figure 1. With the exception of EIEC and EAEC, the other pathotypes were the most widespread among human isolates (Figure 1). Regarding the existence of virulence genes, all the examined isolates harbored *fimH* and *ompA* genes; meanwhile, none of the DEC pathotypes carried the *kpsMTII*, *hly*, or *vt2e* genes. Interestingly, the *astA* gene was detected among all pathotypes with a high prevalence rate among EAEC (Figure 2).

### 3.2. Serotyping of DEC Pathotypes

A total of 119 DEC isolates were serotyped into seven various serotypes, including O26, O45, O55, O151, O125, O145, and O1; meanwhile, 21 isolates (15%) were untypeable. The distribution of the identified serotypes among the recovered DEC isolates is illustrated in Figure 2. All *E. coli* pathotypes were detected among the O55 and O151 serotypes. Of note, all isolates belonging to the O145 serotype were ETEC. The EPEC pathotypes could not be detected among the O45, O145, O1, and O125 serotypes. All isolates of the O26 serotype were distributed among all pathotypes except EAEC (Figure 2).

### 3.3. Antimicrobial Susceptibility Patterns of DEC Pathotypes

Taken together, the analyzed antimicrobial susceptibility data revealed significant variations in the resistance patterns of the investigated DEC isolates against selected antimicrobial agents (*p* > 0.05). Notably, 90% of the tested isolates exhibited the MDR patterns, with additional evidence of the high occurrence of antimicrobial resistance among human isolates. The resistance patterns of different pathotypes are illustrated in Figure 2. Notably, high resistance rates to sulfamethoxazole/trimethoprim, ciprofloxacin, piperacillin/tazobactam, and cefepime and a maximum overall sensitivity to imipenem were observed among the EPEC pathotype. Moreover, high resistance rates against amoxicillin/clavulanic acid, aztreonam, imipenem, erythromycin, and chloramphenicol as well as a lower resistance percentage to tetracycline were noticed among the EIEC pathotype. Of note, the maximum susceptibility rates to amoxicillin/clavulanic acid, cefoxitin, piperacillin/tazobactam, and sulfamethoxazole/trimethoprim were detected among the STEC pathotype. Moreover, high resistance percentages to ampicillin, cefoxitin, cefoperazone, gentamycin, and tetracycline and lower levels of resistance to ampicillin, aztreonam, cefoperazone, cefepime, chloramphenicol, erythromycin, gentamycin, and ciprofloxacin were noticed among the ETEC and EAEC pathotypes, respectively. With the exception of EIEC and STEC, imipenem was the most effective antimicrobial drug against the other *E. coli* pathotypes. Meanwhile, tetracycline and piperacillin/tazobactam were highly effective against EIEC and STEC, respectively. 

### 3.4. Correlation between Pathotypes and Antimicrobial Resistance, Serotypes, Virulence Gene Existence, and Host Types 

A high variability in the antimicrobial resistance patterns was detected across the DEC pathotypes. In general, the antimicrobial resistance was affected by the DEC pathotypes. The sensitive phenotype was correlated with EAEC and STEC; meanwhile, the resistance phenotypes were correlated with other pathotypes (*r*-value = 0:0.5), as shown in Figure 3. Notably, the ETEC pathotype was positively correlated with O26 and O145 (*r*-value = 0.3 for each) but it was negatively correlated with O45 and O125 (*r*-value = −0.2 and −0.1, respectively). Moreover, there were no correlations between ETEC and O1, O55, and O151. Concerning the EPEC pathotype, it had a positive relationship with O26 and O55 (*r*-value = 0.2 for each) and negative correlations with other serotypes (*r*-value = −0.1:−0.3). Positive correlations were announced between EIEC and O125 (*r*-value = 0.1); EAEC and O45 (*r*-value = 0.3) and O1 (*r*-value = 0.2); and STEC and O55, O125, and O45 (*r*-value = 0.1 for each). On the other hand, there were negative correlations between EAEC and O26, O55, and O145 (*r*-value = −0.3, −0.2, and −0.1, respectively) and STEC and O26, O145, and O1 (*r*-value = −0.2, −0.1, and −0.1, respectively). Moreover, there were no correlations between EIEC and O26, O45, O55, O151, and O1 and STEC and O151. As was expected, there were strong positive correlations between the investigated pathotypes and their respective virulence genes (*r*-value = 1). In another context, the diarrheagenic human infections were always associated with the ETEC and EPEC pathotypes; however, both the EIEC and EAEC pathotypes were commonly identified among the recovered animal isolates. Notably, the STEC pathotype was not correlated with human or animal hosts (Figure 3). 

### 3.5. Phenotyping and Molecular Genotyping of DEC Isolates within and among Different Pathotypes

On the basis of serotyping and antimicrobial resistance and virulence gene patterns, our 140 investigated DEC isolates, except six isolates (two pairs of equine and human isolates and one pair of cow and equine isolates), were typed into diverse lineages (Figure 4). The three pairs of the previously related isolates included two EAEC, two EIEC, and one pair of EAEC with EPEC pathotypes (Figure 4). Interestingly, these typing tools showed high discriminatory power (DI-value = 0.999). Complementary to this, another molecular typing tool used in our manuscript was ERIC-PCR. The relationship between DEC isolates within and among different pathotypes can be distinguished on the basis of ERIC banding patterns (Figure 5). The dendrogram constructed on the basis of different amplicons of the ERIC-PCR assay is shown in Figure 6. As demonstrated in the dendrogram, there were 16 branches of related DEC isolates. They comprised 13 branches of two isolates, 2 branches of three isolates, and 1 branch of four isolates. In spite of the presence of these related isolates, this molecular typing tool showed high discriminatory power (DI-value = 0.997). Fortunately, the related isolates within the 13 branches observed in the ERIC-PCR dendrogram were distinguished according to serotyping, and antimicrobial resistance and virulence gene profiles. Moreover, the related six isolates, grouped into three pairs, as illustrated in Figure 4, gave different ERIC-PCR fingerprints. Therefore, it is noteworthy that all our investigated isolates were typed, confirming the high heterogenicity of all DEC, even within each serotype or pathotype. 

## 4. Discussion

The treatment failure crises has been reported for the resistant pathogens, especially in Egypt [30]. Furthermore, the resistant strains, such as *E. coli* as well as *Staphylococcus aureus* and *Salmonella* sp. [31,32,33], harbor several virulence arrays, which increase the morbidity of the diseases caused by these pathogens. Therefore, epidemiological studies of these strains are urgently needed. Interestingly, this report has focused on the detection of genetic relatedness between DEC pathotypes by the ERIC-PCR technique and additionally investigated the correlations between the DEC pathotypes and the antimicrobial resistances, the existence of virulence genes, serotypes, and the types of hosts to develop efficient strategies for the control and treatment of this pathogen. 

Of note, ETEC was the most prevalent pathotype among our DEC isolates and among the isolates investigated in the west of Iran [34]; meanwhile, EPEC and STEC were the most predominant pathotypes in other previous studies [35,36]. The variations in the prevalence of DEC pathotypes may be correlated with environmental contamination and the nutritional types for each geographic area in addition to the reservoir roles of household animals [37]. Regarding the resistance profiles of DEC pathotypes, the prevalence of MDR isolates among the ETEC and EPEC pathotypes, which were mainly associated with the human diseases, was shocking. These results strengthen the hypothesis of the continuous evolution of microbial resistance towards the available antimicrobial drugs. Therefore, the international society strongly advocates the implementation of continuous monitoring strategies for antimicrobial resistance among DEC strains in line with the One Health concept [38].

There are scarce studies investigating the distribution of *E. coli* pathotypes among human and animal hosts. Although previous studies contradict the hypothesis of host specificity and adaptation of DEC pathotypes due to cross reactivity between human and animal [39,40], we found that human and animal infections were associated with the ETEC and EPEC, and EIEC and EAEC pathotypes, respectively. Concerning the correlation between pathotypes and serotypes, there were variations in the DEC serotypes among previous studies [41,42]. This variation could be accepted depending on the variable correlation between *E. coli* pathotypes, serotypes, and the clinical forms of their infections. 

ERIC-PCR is a cost-effective fingerprint and fast molecular typing technique. The clonal variability of DEC can be assessed using this molecular typing tool. In this research, the ERIC-PCR technique showed high discriminatory power (DI-value = 0.997). In accordance with our findings, a high degree of heterogeneity among *E. coli* isolates was observed in previous reports on the basis of the analysis of ERIC-PCR results [43,44]. Therefore, the ERIC-PCR approach has been identified as a good tool to differentiate between the closely related strains. Furthermore, there were large genetic variations among the DEC strains within each pathotype, as the strains of single pathotypes belonged to various phenotypic and genotypic lineages, suggesting that there is no definite factor responsible for DEC resistance and virulence and their expression levels may be affected by numerous conditions.

## 5. Conclusions

Strong positive correlations between DEC pathotypes and the existence of virulence markers and hosts were confirmed; meanwhile, the serotypes could not be correlated well with the DEC pathotypes. Therefore, we suggested that the DEC pathotypes were the powerful factor affecting the degree of morbidity and the choice of treatment protocols. Moreover, our results presented an indication of the heterogeneous nature of DEC pathotypes, as evidenced by the ERIC-PCR results. Our shocking results for the high prevalence of MDR strains and the heterogeneous nature of DEC pathotypes may encourage health care decision makers to introduce more restricted recommendations and surveillance strategies. Additionally, the pathotyping and phenotypic or genotypic testing should be incorporated in medical settings due to the positive correlations between DEC pathotypes and antimicrobial resistance patterns. Therefore, more surveys are required to collect further information about the epidemiological details of DEC isolates to recognize the sources of their infections and implement control measures for avoiding their spread.

## Figures and Tables

**Figure 1 biology-11-01004-f001:**
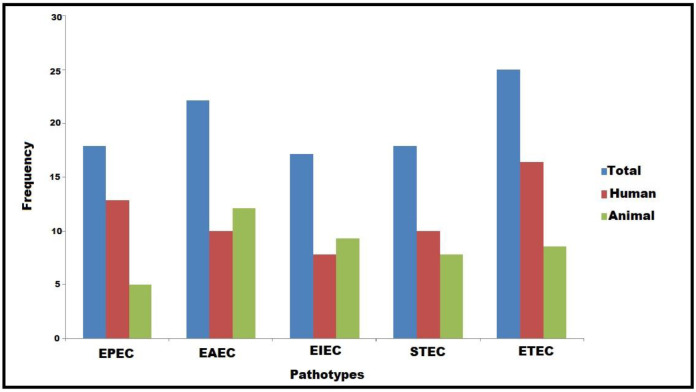
Occurrence and distribution of DEC pathotypes among human and animal hosts. EAEC: enteroaggregative *E. coli*, ETEC: enterotoxigenic *E. coli*, EPEC: enteropathogenic *E. coli*, EIEC: enteroinvasive *E. coli*, STEC: shiga-toxin-producing *E. coli*.

**Figure 2 biology-11-01004-f002:**
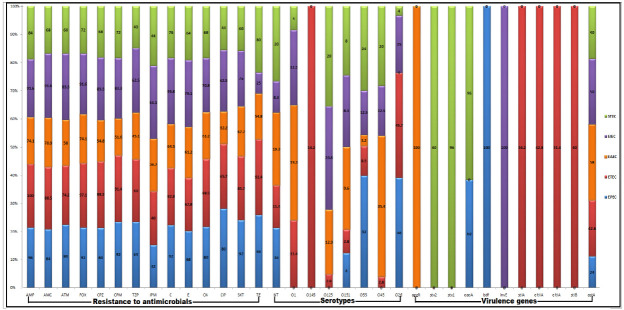
Frequency of resistance to antimicrobials, serotypes, and virulence markers of DEC pathotypes. TZP: piperacillin/tazobactam, C: chloramphenicol, TE: tetracycline, AMP: ampicillin, CN: gentamycin, E: erythromycin, ATM: aztreonam, SXT: sulfamethoxazole/trimethoprim, AMC: amoxicillin/clavulanic acid, IPM: imipenem, CIP: ciprofloxacin, FOX: cefoxitin, CPM: cefepime, CPZ: cefoperazone, EAEC: enteroaggregative *E. coli*, ETEC: enterotoxigenic *E. coli*, EPEC: enteropathogenic *E. coli*, EIEC: enteroinvasive *E. coli*, STEC: shiga-toxin-producing *E. coli*.

**Figure 3 biology-11-01004-f003:**
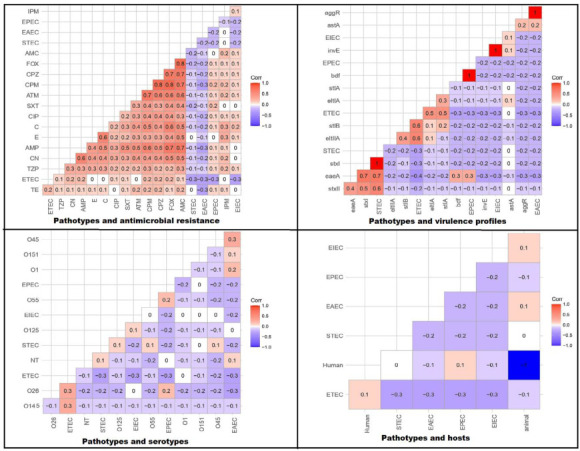
Correlation coefficient (*r*) between diarrheagenic *E. coli* pathotypes and antimicrobial resistance, virulence gene existence, serotypes, and host types. Red and blue colors specify positive and negative correlations, respectively. The color key denotes the correlation coefficient (*R*). The darker red and blue colors indicate stronger positive (*R* = 0.5:1) and negative (*R* = −0.5:−1) correlations, respectively. TZP: piperacillin/tazobactam, TE: tetracycline, AMP: ampicillin, IPM: imipenem, CN: gentamycin, E: erythromycin, ATM: aztreonam, CIP: ciprofloxacin, C: chloramphenicol, SXT: sulfamethoxazole/trimethoprim, AMC: amoxicillin/clavulanic acid, FOX: cefoxitin, CPM: cefepime, CPZ: cefoperazone, EAEC: enteroaggregative *E. coli*, ETEC: enterotoxigenic *E. coli*, EPEC: enteropathogenic *E. coli*, EIEC: enteroinvasive *E. coli*, STEC: shiga-toxin-producing *E. coli*.

**Figure 4 biology-11-01004-f004:**
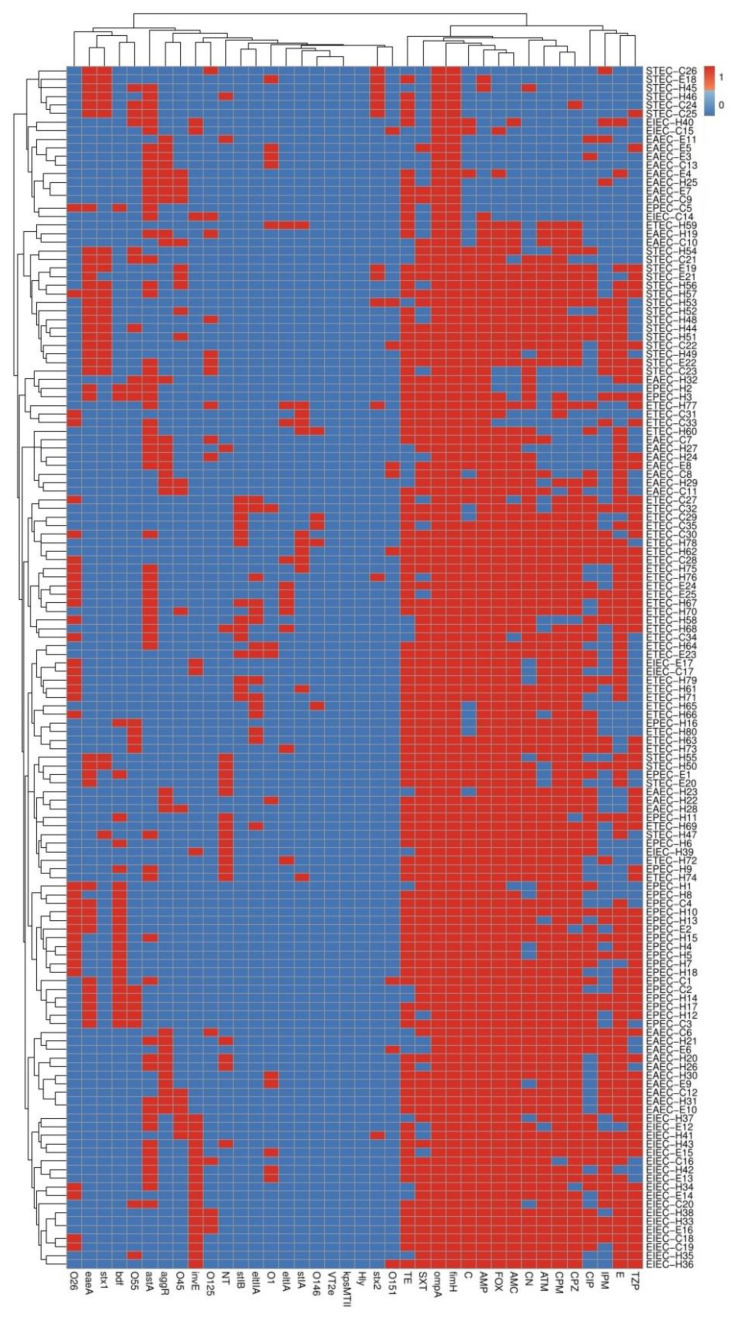
Heat map and hierarchical clustering of diarrheagenic *E. coli* pathotypes according to the occurrence of serotypes, antimicrobial resistances, and virulence genes. Blue and red colors indicate the sensitivity and resistance to a certain antimicrobial and to the absence and presence of a particular serotype and virulence gene, respectively. The code numbers on the right side of the heat map denote the numbers of diarrheagenic *E. coli* pathotypes from equine (E), cow (C), and human (H) sources. TZP: piperacillin/tazobactam, C: chloramphenicol, TE: tetracycline, AMP: ampicillin, CN: gentamycin, E: erythromycin, ATM: aztreonam, SXT: sulfamethoxazole/trimethoprim, AMC: amoxicillin/clavulanic acid, IPM: imipenem, CIP: ciprofloxacin, FOX: cefoxitin, CPM: cefepime, CPZ: cefoperazone, EAEC: enteroaggregative *E. coli*, ETEC: enterotoxigenic *E. coli*, EPEC: enteropathogenic *E. coli*, EIEC: enteroinvasive *E. coli*, STEC: shiga-toxin-producing *E. coli*.

**Figure 5 biology-11-01004-f005:**
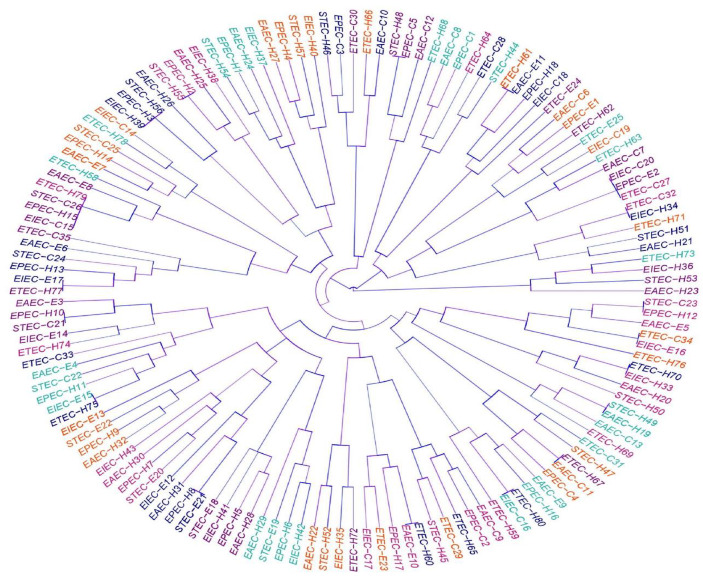
Dendrogram showing the relatedness of diarrheagenic *E. coli* pathotypes isolated from equine (E), cow (C), and human (H) origins, as determined by ERIC-PCR fingerprinting. EAEC: enteroaggregative *E. coli*, ETEC: enterotoxigenic *E. coli*, EPEC: enteropathogenic *E. coli*, EIEC: enteroinvasive *E. coli*, STEC: shiga-toxin-producing *E. coli*.

**Figure 6 biology-11-01004-f006:**
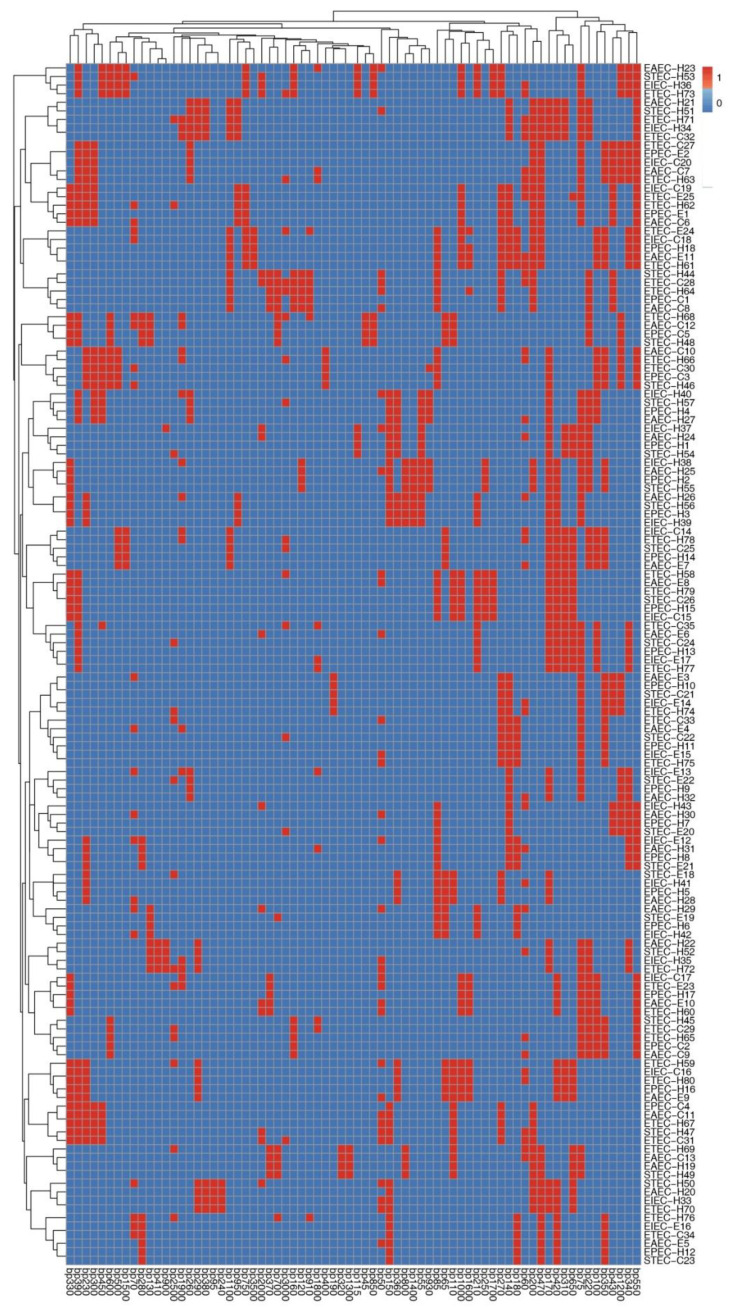
Heat map and hierarchical clustering of diarrheagenic *E. coli* pathotypes, according to the generated ERIC-PCR amplicon size (bp) profiles. Blue and red colors indicate the absence and presence of particular ERIC-PCR bands, respectively. The code numbers on the right side of the heat map denote the numbers of diarrheagenic *E. coli* pathotypes from equine (E), cow (C), and human (H) sources. TZP: piperacillin/tazobactam, C: chloramphenicol, TE: tetracycline, AMP: ampicillin, CN: gentamycin, E: erythromycin, ATM: aztreonam, SXT: sulfamethoxazole/trimethoprim, AMC: amoxicillin/clavulanic acid, IPM: imipenem, CIP: ciprofloxacin, FOX: cefoxitin, CPM: cefepime, CPZ: cefoperazone, EAEC: enteroaggregative *E. coli*, ETEC: enterotoxigenic *E. coli*, EPEC: enteropathogenic *E. coli*, EIEC: enteroinvasive *E. coli*, STEC: shiga-toxin-producing *E. coli*.

**Table 1 biology-11-01004-t001:** The primer sequences and the predicted amplicon sizes of the diarrheagenic *E. coli* genes under study.

Target Gene	Specificity	Primer Sequence (5′-3′)	Amplicon Size (bp)	Reference
*16S rRNA*	RNA component of the 30S ribosomal subunit	F: GACCTCGGTTTAGTTCACAGA R: CACACGCTGACGCTGACCA	585	[12]
*ompA*	Outer membrane protein	F: AGCTATCGCGATTGCAGTG R: GGTGTTGCCAGTAACCGG	919	[18]
*kpsMTII*	Adhesion	F: CAGGTAGCGTCGAACTGTA R: CATCCAGACGATAAGCATGAGCA	280	[18]
*hly*	Hemolysin	F: AACAAGGATAAGCACTGTTCTGGCT R: ACCATATAAGCGGTCATTCCCGTCA	117	[15]
*stx2*	Shiga toxin 2	F: CCATGACAACGGACAGCAGTT R: CCTGTCAACTGAGCAGCACTTTG	779	[17]
*stx1*	Shiga toxin 1	F: ACACTGGATGATCTCAGTGG R: CTGAATCCCCCTCCATTATG	614	[17]
*fimH*	Adhesion	F: TGCAGAACGGATAAGCCGTGG R: GCAGTCACCTGCCCTCCGGTA	508	[21]
*vt2e*	Vero toxin	F: CCTTAACTAAAAGGAATATA R: CTGGTGGTGTATGATTAATA	230	[14]
*astA*	Enterotoxin	F: TGCCATCAACACAGTATATCC R: TCAGGTCGCGAGTGACGGC	116	[16]
*invE*	Transcriptional regulation of invasion	F: CGATAGATGGCGAGAAATTATATCCCG R: CGATCAAGAATCCCTAACAGAAGAATCAC	766	[22]
*aggR*	Transcriptional activator of adherence fimbriae	F: ACGCAGAGTTGCCTGATAAAG R: AATACAGAATCGTCAGCATCAGC	400	[19]
*eaeA*	Intimin	F: GTAAAGTCCGTTACCCCAACCTG R: GCACACGGAGCTCCTCAGTCTCC	218	[24]
*eltIA*	Heat-labile toxin I	F: TTACGGCGTTACTATCCTCTCTA R: GGTCTCGGTCAGATATGTGATTC	275	[20]
*eltIIA*	Heat-labile toxin II	F: ATATCATTTTCTGTTTCAGCAAA R: CAATAAAATCATCTTCGCTCATG	720	[20]
*stIA*	Heat-stable toxin A	F: TTTCCCCTCTTTTAGTCAGTCAA R: GCAGGATTACAACACAATTCACAGCAG	159	[25]
*stIB*	Heat-stable toxin B	F: TGCTAAACCAGTAGAGTCTTCAAAA R: GCAGGATTACAACACAATTCACAGC	138	[25]
*bfp*	Bundle-forming pilus	F: GACACCTCATTGCTGAAGTCG R: CCAGAACACCTCCGTTATGC	910	[19]

**Table 2 biology-11-01004-t002:** Zone diameter and MIC breakpoints for diarrheagenic *E. coli* isolates.

Antimicrobial Agent	Symbol	Conc. (ug)	Interpretative Categories			
			Zone Diameter Breakpoints (mm)		MIC Breakpoints (μg/mL)	
			Resistance	Sensitive	Resistance	Sensitive
Ampicillin	AMP	10	≤13	≥17	≥32	≤8
Amoxicillin/clavulanic acid	AMC	20/10	≤13	≥18	≥32/16	≤8/4
Aztreonam	ATM	30	≤17	≥21	≥16	≤4
Cefoxitin	FOX	30	≤14	≥18	≥32	≤8
Cefoperazone	CPZ	75	≤15	≥21	≥64	≤16
Cefepime	CPM	30	≤18	≥25	≥16	≤2
Piperacillin/tazobactam	TZP	100/10	≤17	≥21	≥128/4	≤16/4
Chloramphenicol	C	30	≤12	≥18	≥32	≤8
Imipenem	IPM	10	≤19	≥23	≥4	≤1
Erythromycin	E	15				
Gentamycin	CN	10	≤12	≥15	≥16	≤4
Ciprofloxacin	CIP	5	≤15	≥21	≥4	≤1
Sulfamethoxazole/trimethoprim	SXT	23.75/1.25	≤10	≥16	≥4/76	≤2/38
Tetracycline	TE	30	≤11	≥15	≥16	≤4

## Data Availability

All data generated or analyzed during this study are included in the submitted manuscript.

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
