# Peer review of "What Is behind the Correlation Analysis of Diarrheagenic E. coli Pathotypes?"

_biology, 2022, doi:10.3390/biology11071004_

Round 1

Reviewer 1 Report

  1. The described study lacks novelty and is of local or regional importance
  2. The manuscript contributes only incremental knowledge to the field of research
  3. The manuscript contains a great deal of recycled information and or hypothesis from other publications of the authors aside from great deal of self citations which must be minimized for your future publication efforts

Author Response

Dear Professor Doctor/ Reviewer 1

The manuscript ID: biology-1708150

Title: What is behind the correlation analysis of diarrheagenic E. coli pathotypes?

Many thanks for all your efforts to improve our manuscript. We would like to thank the reviewer for its raised and thorough comments. The corrections requested by the reviewer have been done point by point as shown in the revision form. All changes in the revised manuscript were highlighted or were done using track change. An English edition for our revised manuscript was done by native English speaker. Hopefully, our revised manuscript meets the expectations of you and the reviewers and be considered for publication in Biology Journal.

1-The described study lacks novelty and is of local or regional importance

Response to reviewer comment:

We totally respect the reviewer criticism. The results of this report was implemented in the infection control guidelines and gave alarming sign for the heterogeneity, the evolution and the wide spread of multidrug resistant side by side with the multi-virulent DEC strains. Additionally from the correlation analysis, the strong positive correlations between DEC pathotypes and the existence of virulence markers and hosts were confirmed; meanwhile, the serotypes could not be well correlated with DEC pathotypes. These results are of great clinical importance for the physicians during the choice of the treatment protocols and also to predict the prognosis of the cases. From our point of view, the results of this manuscript showed clinical novelty and it may enhance the health care decision-makers to put more restricted recommendations and surveillance strategies and these issues were more elucidated in the revised version of our manuscript.

In this study, the DEC clinical strains were isolated from different hosts, hospitals and provinces. The health hazards of DEC are worldwide; the information gained from the studies in certain countries can lay foundation for the infection control strategies throughout the world for this resistant pathogen. Several published reports put solid conclusions for the strains, which were recovered in certain countries.

2-The manuscript contributes only incremental knowledge to the field of research

Response to reviewer comment:

Thank you for this comment. Up to our knowledge, this is among the first reports, which gave interesting data about the correlation analysis between DEC pathotypes and  antimicrobial resistance, virulence profile, serotypes and host and we obtained solid conclusions for these correlation analysis. Additionally, these results were more elaborated in the revised version of our manuscript

3-The manuscript contains a great deal of recycled information and or hypothesis from other publications of the authors aside from great deal of self-citations, which must be minimized for your future publication efforts

Response to reviewer comment:

Thanks for this insight criticism about our recycled information. Although these hypotheses were related to the outcome of our results in this manuscript, we removed most of this information and reduced the self-citations. 

Reviewer 2 Report

The article of Bendary et al.,   concerns the correlation with the host, resistance to antibiotics, virulence gene profile and serotype of 140 multidrug resistant diarrheagenic E. coli strains previously isolated from humans and animals. The study represents a considerable amount of lab work, is well written with proper methodology and a clear aim. Given the increasing concern of multidrug resistant strains, the study and its conclusions are of interest either for clinicians or for scientists in the field of food safety. However, to present a scientifically sound article, the authors should address some major issues presented below:

Lines 82-84: "Additionally, the correlation analysis between DEC pathogens and antimicrobial resistance as well as virulence profiles may help the physicians to avoid the treatment failure”. As it is not so simple, please elaborate on this. Any relative refs?

Line 101: Where is table S1. There is no supplementary material in submission files.

Section 2.5 and Table 2. Please add the concentration of the antibiotics tested and related refs for the MIC breakpoints.

Section 2.7. In this section, please add/describe the correlation coefficient estimation.

Lines 228,231. You are using expressions like “unreasonably” or “notably” when you are referring to you results. Why they are “unreasonable” or “notable”?

Lines 242-243, “With the exception of EIEC and STEC, 242 imipenem was the drug of choice for treating other E. coli pathotypes”. Why? Was it the most effective? Is this sufficient for choosing this particular drug?

In statistics, correlation coefficients lower than 0.75 aren’t considered as “strong positive” and R=0.5 is an indication of a weak positive correlation. Therefore, I’m recommending a) to add the values of the coefficients in the text at least when “strong” relationships are discussed (paragraph 3.4, lines 261-262) and accordingly b) to extend the spectrum of color coding and include the values ±0.75 in figure 3.

Line 421, Conclusions. “Strong positive correlations between…”. Please revise accordingly based on the above.

References. I believe that there are too many self-citations.

Author Response

Dear Professor Doctor/ Reviewer 2

The manuscript ID: biology-1708150

Title: What is behind the correlation analysis of diarrheagenic E. coli pathotypes?

Many thanks for all your efforts to improve our manuscript. We would like to thank the reviewer for its raised and thorough comments. The corrections requested by the reviewer have been done point by point as shown in the revision form. All changes in the revised manuscript were highlighted or were done using track change. An English edition for our revised manuscript was done by native English speaker. Hopefully, our revised manuscript meets the expectations of you and the reviewers and be considered for publication in Biology Journal.

The article of Bendary et al.,   concerns the correlation with the host, resistance to antibiotics, virulence gene profile and serotype of 140 multidrug resistant diarrheagenic E. coli strains previously isolated from humans and animals. The study represents a considerable amount of lab work, is well written with proper methodology and a clear aim. Given the increasing concern of multidrug resistant strains, the study and its conclusions are of interest either for clinicians or for scientists in the field of food safety. However, to present a scientifically sound article, the authors should address some major issues presented below:

1-Lines 82-84: "Additionally, the correlation analysis between DEC pathogens and antimicrobial resistance as well as virulence profiles may help the physicians to avoid the treatment failure”. As it is not so simple, please elaborate on this. Any relative refs?

Response to reviewer comment:

Thank you for your comment. It was adjusted according to your recommendation.

2-Line 101: Where is table S1. There is no supplementary material in submission files.

Response to reviewer comment:

Thank you for your excellent revision. We apologized for this mistake. Actually, there is no supplementary Table, but we meant Table 1 not Table S1 and this was corrected in this revised version of our manuscript.

3-Section 2.5 and Table 2. Please add the concentration of the antibiotics tested and related refs for the MIC breakpoints.

Response to reviewer comment:

Thank you for your comment. We added the concentration of the tested antimicrobial agents and the related references.

4-Section 2.7. In this section, please add/describe the correlation coefficient estimation.

Response to reviewer comment:

Thank you for your comment. It was described

5-Lines 228, 231. You are using expressions likeunreasonably” or notablywhen you are referring to you results. Why they are “unreasonable” or “notable”?

Response to reviewer comment:

Thank you for your comment. The term “unreasonably” was used to reflect the unexpected results, which were surprised. The prevalence of MDR isolates was very high in this report (90%). On the other hand, the term “notable” was used to reflect an important observation such as the high resistance rate to certain antimicrobial agents. Therefore, we try to highlight these results using the terms “unreasonably” andnotable”.

6-Lines 242-243, With the exception of EIEC and STEC, 242 imipenem was the drug of choice for treating other coli pathotypes”. Why? Was it the most effective? Is this sufficient for choosing this particular drug?

Response to reviewer comment:

Thank you for this comment. According to our antimicrobial sensitivity results, imipenem was the most effective drugs against all DEC pathotypes except EIEC and STEC and to avoid the confusion, we replaced the term “drug of choice” to “the most effective drug”.

7-In statistics, correlation coefficients lower than 0.75 aren’t considered as “strong positive” and R=0.5 is an indication of a weak positive correlation. Therefore, I’m recommending a) to add the values of the coefficients in the text at least when “strong” relationships are discussed (paragraph 3.4, lines 261-262) and accordingly b) to extend the spectrum of color coding and include the values ±0.75 in figure 3.

Response to reviewer comment:

Thank you for this comment. It was revised and the values were added according to the reviewer recommendations.

8-Line 421, Conclusions. Strong positive correlations between”. Please revise accordingly based on the above.

Response to reviewer comment:

Thank you for this comment. It was revised and confirmed.

9-As it was expected according to the information in the literature, each DEC pathotypes were harbored with certain virulence upon, which these pathotypes were categorized.

Response to reviewer comment:

These were confirmed in our manuscript. Strong positive correlations between DEC pathotypes and the existence of virulence markers and hosts were confirmed.

10- I believe that there are too many self-citations.

Response to reviewer comment:

Thanks for this insight criticism about our self-citations. Although these citations were related to the outcome of our results in this manuscript, we removed most of these self-citations in the revised version of the manuscript. 

Round 2

Reviewer 1 Report

1- Figure 2 has poor quality, too crowded, and not readable, it must be represented in a better way

2- Optimize the color scale for the heat maps in figures 4 and 6

3- Several references are not relevant to the topic of research

Author Response

Dear Professor Doctor/ reviewer 1

The manuscript ID: biology-1708150

Title: What is behind the correlation analysis of diarrheagenic E. coli pathotypes?

Many thanks for all your efforts to improve our manuscript. We would like to thank the reviewer for its raised and thorough comments. The corrections requested by the reviewer have been done point by point as shown in the revision form. All changes in the revised manuscript were highlighted or were done using track change. An English edition for our revised manuscript was done by native English speaker. Hopefully, our revised manuscript meets the expectations of you and the reviewers and be considered for publication in Biology Journal.

Response to the comments:

Reviewer 1:

  • Figure 2 has poor quality, too crowded, and not readable, it must be represented in a better way

Response to reviewer comment:

Thank you for your comments; we change the design of this figure to other better readable type

  • Optimize the color scale for the heat maps in figures 4 and 6

Response to reviewer comment:

Thank you for the excellent reviewing, we change and optimize the color code in these figure 

  • Several references are not relevant to the topic of research

Response to reviewer comment:

Thank you for your comments; All references were double checked and we modified some references in accordance to this comments

Reviewer 2 Report

I believe that the manuscript has been considerably improved.

Author Response

Dear Professor Doctor/ Reviewer 2

The manuscript ID: biology-1708150

Title: What is behind the correlation analysis of diarrheagenic E. coli pathotypes?

Many thanks for all your efforts to improve our manuscript. We would like to thank the reviewer for its raised and thorough comments. The corrections requested by the reviewer have been done point by point as shown in the revision form. All changes in the revised manuscript were highlighted or were done using track change. An English edition for our revised manuscript was done by native English speaker. Hopefully, our revised manuscript meets the expectations of you and the reviewers and be considered for publication in Biology Journal.

Reviewer 2:

I believe that the manuscript has been considerably improved.

Thank you for your comment that give us positive energy

Round 3

Reviewer 1 Report

I could detect a great deal of plagiarism and data recycling in the result section from this article "Comparative Analysis of Human and Animal E. coli: Serotyping, Antimicrobial Resistance, and Virulence Gene Profiling" published for the same authors in Antibiotics doi: 10.3390/antibiotics11050552

for example this paragraph

"Based on serotyping and antimicrobial resistance, and virulence genes` profiles, all the tested DEC isolates (140) with the exception of six isolates (two pairs of human and equine isolates and one pair of equine and cow isolates) could be typed into different lineages as shown in Figure 4." lines 314-317 page 11 of this manuscript

I can see the same paragraph with modifying the numbers in the published paper

"Based on serotyping, antimicrobial resistance and virulence gene profiles, all the tested DEC isolates (140), with the exception of 14 isolates (3 pairs of human and equine isolates, 2 pairs of human isolates, and 2 pairs of human and cow isolates), could be typed into different lineages, as shown in Figure 4."

Author Response

I could detect a great deal of plagiarism and data recycling in the result section from this article "Comparative Analysis of Human and Animal E. coli: Serotyping, Antimicrobial Resistance, and Virulence Gene Profiling" published for the same authors in Antibiotics doi: 10.3390/antibiotics11050552

Response to reviewer:

thank you for your comment: there is no data recycling, only the two works were performed on the same isolates (140 DEC ) with different techniques and goals as it was referred in the line 104 of  this manuscript. “All isolates were kindly provided from microbiology laboratories of Faculty of Pharmacy, Port-Said University and Faculty of Veterinary Medicine, Zagazig University". These isolates (140) were the start for two separate works,  one which deals with the difference between the phenotypic and genotypic character between human and animal isolates (this work was published in n Antibiotics doi: 10.3390/antibiotics11050552) and the other work deals with the different pathotypes of these isolates and there correlations. In this work we used different virulence factors and different technique than our published paper as ERIC PCR. in accordance to your comment, we remove any plagiarism or similar writing throughout our manuscript. Kindly, carefully read the two paper after our modifications 

thank you in advance